# Infrared and Visible Image Fusion Algorithm Based on Double-Domain Transform Filter and Contrast Transform Feature Extraction

**DOI:** 10.3390/s24123949

**Published:** 2024-06-18

**Authors:** Xu Ma, Tianqi Li, Jun Deng, Tong Li, Jiahao Li, Chi Chang, Rui Wang, Guoliang Li, Tianrui Qi, Shuai Hao

**Affiliations:** 1College of Safety Science and Engineering, Xi’an University of Science and Technology, Xi’an 710054, China; maxu@xust.edu.cn; 2College of Electrical and Control Engineering, Xi’an University of Science and Technology, Xi’an 710054, China; altq8792@163.com (T.L.); litong20221120@163.com (T.L.); 18406060425@stu.xust.edu.cn (J.L.); 23206223096@stu.xust.edu.cn (C.C.); 23206232146@stu.xust.edu.cn (R.W.); 23206232117@stu.xust.edu.cn (G.L.); 20406100226@stu.xust.edu.cn (T.Q.); haoxust@163.com (S.H.)

**Keywords:** high-pass filter, logical filter, feature extraction, nonlinear contrast transform function, color correction

## Abstract

Current challenges in visible and infrared image fusion include color information distortion, texture detail loss, and target edge blur. To address these issues, a fusion algorithm based on double-domain transform filter and nonlinear contrast transform feature extraction (DDCTFuse) is proposed. First, for the problem of incomplete detail extraction that exists in the traditional transform domain image decomposition, an adaptive high-pass filter is proposed to decompose images into high-frequency and low-frequency portions. Second, in order to address the issue of fuzzy fusion target caused by contrast loss during the fusion process, a novel feature extraction algorithm is devised based on a novel nonlinear transform function. Finally, the fusion results are optimized and color-corrected by our proposed spatial-domain logical filter, in order to solve the color loss and edge blur generated in the fusion process. To validate the benefits of the proposed algorithm, nine classical algorithms are compared on the LLVIP, MSRS, INO, and Roadscene datasets. The results of these experiments indicate that the proposed fusion algorithm exhibits distinct targets, provides comprehensive scene information, and offers significant image contrast.

## 1. Introduction

In the field of image fusion, infrared and visible image fusion plays a crucial role and finds extensive applications [1,2]. Within the military domain [3,4], it facilitates target identification, night vision capabilities, and navigation systems. In terms of security applications [5], it enhances nighttime or complex-environment surveillance by providing clearer images to aid in detecting abnormal behavior. The medical sector [6] benefits from its usage for disease diagnosis and treatment monitoring purposes. By employing a specific fusion method to combine infrared and visible images effectively, this technique ensures that the resulting fused image encompasses feature information from both sources while presenting a more comprehensive and accurate depiction [7]. So far, the most widely used method in the field of image fusion is based on a pixel-level approach. This method is broadly categorized into the following four types: spatial-domain [8], transform-domain [9], low-rank matrix [2], and bionic algorithms [10].

Spatial-domain image fusion algorithms that are widely used include but are not limited to weighted average [11], contrast transform [12], and logical filtering algorithms [13]. Spatial-domain image fusion algorithms operate directly on the pixel values of the image, which is simple, intuitive, and easy to understand and has better real-time performance.

Transform-domain image fusion algorithms convert images from their time domain to the frequency domain. They fall into two main groups: frequency-domain filters and multiscale transform techniques. Frequency-domain filtering methods excel in feature extraction, capturing more image details and texture features compared to spatial-domain fusion algorithms. Multiscale transformation is widely recognized as an effective technique for achieving image fusion. The image fusion algorithm based on a multiscale transform can be divided into three major steps: multiscale decomposition, fusion rules, and inverse multiscale transform [14]. Currently, widely used multiscale transform fusion algorithms can be categorized into two main groups according to the transform rules: pyramid transforms and wavelet transforms. A pyramid transform contains contrast pyramids, Laplace pyramid transform, and so on. In the field of image fusion based on the wavelet transform, Pu et al. [15] proposed an image fusion algorithm based on contrast transformation and wavelet decomposition. In addition, Li et al. [16] proposed an image fusion method combined with morphological image enhancement and a dual-tree complex wavelet. This method was a good solution for images with ringing, incomplete scene information, low contrast, and Gibbs artifacts. Li et al. [17] introduced an image fusion scheme based on NSCT and low-level visual features. Moreover, Tan et al. [18] proposed an image fusion via NSST and PCNN [19] in the multiscale morphological gradient (MSMG) domain and explored a new fusion method in the MSMG domain. On the basis of NSST, Li et al. [20] proposed the LSWT-NSST image fusion algorithm, which effectively solved edge blurring and Gibbs phenomena in the traditional wavelet transform algorithm and the loss of subtle features in the NSST.

Low-rank matrix algorithms can be broadly classified into three groups depending on the techniques employed: matrix decomposition algorithms, matrix representation algorithms, and approaches utilizing low-rank kernel approximation. Matrix decomposition methods work by stacking multiple images into a matrix and then performing a low-rank decomposition of that matrix. Suryanarayana et al. [21] designed the multiple degradation skilled network for infrared and visible image fusion based on a multi-resolution SVD update, effectively improving the resolution of the fused image. Principal Component Analysis (PCA) [22] has desirable denoising effect. It has no parameter limit in the fusion process, and the calculation is small. Wang et al. [23] presented an improved image fusion method based on NSCT and accelerated Non-negative Matrix Factorization (NMF) and obtained better fusion results than with PCA. Low-rank representation methods are based on the assumption that the information of an image can be represented by a low-rank matrix. Image fusion can be achieved by decomposing each image into a combination of a low-rank matrix and a sparse matrix and then performing a weighted summation on the low-rank part. By combining a Low Rank Representation (LRR) and a Sparse Representation (SR), Wang et al. [24] proposed a low-rank sparse representation (LSRS) to provide new theoretical knowledge for image fusion. Low-rank kernel approximation-based methods perform image fusion by transforming the image convolution operation into a low-rank influence. Abdolali et al. [25] introduced a multiscale decomposition in the low-rank approximation in their paper and provided a theoretical basis for a Multiscale Low-Rank Approximation (MLRA) which was then applied to image processing. Low-rank matrix algorithms can have disadvantages such as high computational complexity, poor noise immunity, and the need for prior knowledge in some cases. When applying these algorithms, it is crucial to consider their limitations. To achieve optimal fusion results, we must carefully select appropriate parameters and make adjustments to the algorithm design based on the specific situation.

An important branch of bionic algorithms is the deep learning algorithms based on neural networks, which typically enforce desired distributional properties in the fused images by constructing an objective function. Convolutional Neural Networks (CNNs) [26] have gained popularity in image fusion due to their powerful feature extraction capabilities. Deep learning approaches usually adopt an end-to-end strategy for fusion, minimizing the need for preprocessing, parameter tuning, and post-processing. However, there are limited training data available for deep learning methods in image fusion compared to tasks like target recognition and detection, which affects the performance of deep learning models.

Currently, infrared and colorful visible images are a popular research topic with diverse applications in daily life. However, two issues arise during the fusion process: a loss of texture details due to color information loss and edge blurring of the infrared target. To address these problems, we propose a new algorithm called DDCTFuse that is based on a double-domain transform filter and nonlinear transform feature extraction. Our specific contributions are listed below:

(1) In order to improve the quality and flexibility of images in the frequency-domain decomposition, an adaptive high-pass filtering algorithm is proposed and used in the frequency-domain decomposition. Meanwhile, the high-frequency texture is denoised and the edge features are preserved by introducing bilateral filtering.

(2) Aiming at the problem that the infrared target of fusion image is not obvious, in this paper, a novel nonlinear element transform function is designed, and a new feature extraction algorithm for infrared targets is proposed. The algorithm is very sensitive to the high-energy part of infrared images and can significantly enhance and extract infrared targets.

(3) A novel spatial-domain logical filter image optimization strategy is proposed, which effectively decreases “virtual shadow” (artifacts) and preserves fine details caused by the Gibbs phenomenon (ringing effect) in the frequency-domain fusion process.

The remaining sections of the paper are structured as follows: In Section 2, we provide a concise overview of the bilateral filter and establish the theoretical foundation for the subsequent sections. In Section 3, we describe the process and principle of DDCTFuse in detail. Experimental details, parameter settings, and result analysis are provided in Section 4. Finally, results are discussed in Section 5 and summarized in Section 6.

## 2. Bilateral Filter

The bilateral filter (BF) is a well-known edge-preserving filter and a classical nonlinear spatial filtering method. The bilateral filter integrates the effects of the spatial and value domains on the filtering generation; thus, it can achieve the effect of noise reduction and smoothing and is a good edge-preserving filter. The bilateral filter was employed in this paper to effectively attenuate high-frequency details while preserving the intricate texture, thereby achieving high-frequency denoising.

A bilateral filter is realized by a weighted average method based on a Gaussian distribution. The principle of the bilateral filter implementation is illustrated by solving specific pixel values in the image as an example. The output pixel value g of the image at (i,j) after bilateral filtering depends on the weighted combination of pixel values *f* in the neighborhood.
(1)$$P\left(i,j\right)=\frac{\sum_{k,l}F\left(k,l\right)E\left(i,j,k,l\right)}{\sum_{k,l}E\left(i,j,k,l\right)}$$
where *k* and *l* denote the position of the neighboring pixel, and the weight coefficient E(i,j,k,l) is the product of the spatial-domain kernel *d* and the value-domain kernel *r*. The spatial-domain kernel *d* is the Euclidean distance between the current point and the center point calculated based on the Gaussian function.
(2)$$Y\left(i,j,k,l\right)=\exp\left(-\frac{{\left(i-k\right)}^{2}+{\left(j-l\right)}^{2}}{2\sigma_{Y}^{2}}\right)$$

The value-domain kernel *r* is the absolute value of the difference between the current point and the center pixel value based on a Gaussian function.
(3)$$T\left(i,j,k,l\right)=\exp\left(-\frac{{∥f\left(i,j\right)-f\left(k,l\right)∥}^{2}}{2\sigma_{T}^{2}}\right)$$

From the formulae for the space-domain kernel and the value-domain kernel, the weight coefficients can be calculated as:(4)$$E\left(i,j,k,l\right)=\exp\left(-\frac{{\left(i-k\right)}^{2}+{\left(j-l\right)}^{2}}{2\sigma_{Y}^{2}}-\frac{{∥f\left(i,j\right)-f\left(k,l\right)∥}^{2}}{2\sigma_{T}^{2}}\right)$$

In order to more intuitively reflect the effect of the algorithm, the specific details of the enhanced output results are shown in Figure 1.

## 3. The Proposed DDCTFuse Method

In this section, we provide a detailed introduction of DDCTFuse, as shown in Figure 2. Also, the mathematical principles of DDCTFuse are presented in this section.

The YCbCr color space transform has the advantages of being better at processing image brightness and color information, reducing data redundancy, and improving compression efficiency. As a result, the Y-channel information of the RGB visible image was extracted using the YCbCr transform. Right after that, the Y-channel image (Y) and infrared image (IR) were decomposed into high-frequency and low-frequency components through the adaptive high-pass filter. Then, the extracted high-frequency detail features (including noise) were processed using BF, preserving the detail texture while reducing the noise. Meanwhile, the nonlinear contrast transform feature extraction was performed on the IR image to obtain a feature map, which was used to guide the fusion of the Y-channel image and the high-frequency part of the IR image. The low-frequency part was fused by wavelet weighting to obtain the final base layer. Afterwards, a spatial logic filter optimization strategy was devised to enhance the infrared target while preserving the color and texture information of the source image, effectively optimizing the quality of the fused image and obtaining the final result. Finally, an inverse YCbCr transform generated the final fusion result.

### 3.1. Adaptive High Pass Filter

The high-frequency component of an image represents sudden changes in intensity, which can correspond to edges or noise, and often both. The low-frequency component represents the portion of the image that undergoes gradual changes, specifically referring to the contour information within the image.

In traditional Butterworth filtering, the cutoff frequency is usually set to a certain value and applied to all images. This is not conducive to the extraction of high-frequency details, which often leads to the incomplete extraction of high-frequency details or high-frequency information containing low-frequency components, which reduces the robustness of the algorithm.

In order to solve the above problems, we designed an adaptive high-pass filter and dynamically adjusted the cutoff frequencies. The mean frequency of the image can be obtained by averaging the amplitude. We took the mean frequency as the cutoff frequency of the high-pass filter to preserve more detail in the image. The specific mathematical expression is given below.

First, we performed a domain transformation of the image. The Fourier expression of the image is discontinuous and its discrete expression is:(5)$$F\left(u,\nu \right)=\frac{1}{MN}\sum_{x=0}^{M-1}\sum_{y=0}^{N-1}f\left(x,y\right)e^{-j2\pi (\frac{ux}{M}+\frac{vy}{N})}$$
where u=0,1,2,…,M−1; v=0,1,2,…,N−1; f(x,y) denotes the grayscale value of the point; *M* and *N* denote the length and width of the image; F(u,v) denotes the Fourier transform result of the image.

Second, the cutoff frequency was calculated. In order to find the mean frequency of the image, the Fourier transform of the image was transformed into a polar coordinate representation.
(6)$$F\left(u,v\right)=\left|\begin{array}{c} F(u,v) \end{array}\right|e^{-j\phi (u,v)}$$
where $\left|F(u,v)\right|$ denotes the magnitude and ϕ denotes the phase angle.

The expression for the magnitude is:(7)$$\left|F\left(u,v\right)\right|={\left[\begin{array}{c} R{\left(u,v\right)}^{2}+I{\left(u,v\right)}^{2} \end{array}\right]}^{\frac{1}{2}}$$
where R(u,v) and I(u,v) denote the real and imaginary parts of F(u,v).

Then, the frequency coordinates corresponding to the average amplitude were calculated, and their mathematical expressions are shown below:(8)$$A_{0}=\frac{1}{M*N}\sum_{u=0}^{M-1}\sum_{v=0}^{N-1}\left|F\left(u,v\right)\right|$$
(9)$$\left(u_{0},v_{0}\right)=\underset{\left(u,v\right)}{\arg\min}\left|F\left(u,v\right)-A_{0}\right|$$
where $A_{0}$ is the average amplitude of the image, and $\left(u_{0},v_{0}\right)$ is the frequency coordinate that minimizes the difference between the amplitude spectrum F(u,v) and $A_{0}$.

The frequency values corresponding to the frequency coordinates $\left(u_{0},v_{0}\right)$ can be obtained by the following formula:(10)$$f_{u0}=u_{0}\cdot \Delta f_{u}=\frac{B_{h}}{M}$$
(11)$$f_{v0}=v_{0}\cdot \Delta f_{v}=\frac{B_{v}}{N}$$
(12)$$F_{0}=\frac{\left(f_{v0}+f_{u0}\right)}{2}$$
where $f_{v0}$ and $f_{u0}$ represent the horizontal and vertical components of the frequency domain, respectively, $B_{h}$ and $B_{v}$ represent horizontal and vertical bandwidths, and $F_{0}$ is the average frequency.

According to the mathematical formula of the Butterworth high-pass filter, we propose the following formula for the adaptive high-pass filter introduced in this paper:(13)$$H\left(u,\nu \right)=\frac{1}{1+{\left[F_{0}/D\left(u,\nu \right)\right]}^{2n}}$$

Third, a frequency-domain decomposition was performed. The low-frequency part was obtained by performing a two-dimensional Fourier inverse transform of *H* into the time domain (*h*) and subtracting *h* (the high-frequency part) from *f*.
(14)$$h\left(x,y\right)=\frac{1}{M*N}\sum_{u=0}^{M-1}\sum_{v=0}^{N-1}H\left(u,v\right)e^{-j2\pi (\frac{ux}{M}+\frac{vy}{N})}$$
(15)$$l\left(x,y\right)=\sum_{x=0}^{M-1}\sum_{y=0}^{N-1}\left[f\left(x,y\right)-h\left(x,y\right)\right]$$
where h(x,y) denotes the high-frequency part pixel value, and l(x,y) denotes the low-frequency part pixel value.

### 3.2. Contrast Transform

We propose a new nonlinear element transform function to obtain the image contrast conversion coefficient CT and weight the image pixel value to transform the image contrast.

First, the optimal threshold “*G*” was obtained by the Nobuyuki Otsu method. A summary of the pixels in the light (*B*) and dark (*D*) parts of the image was obtained by comparing and counting the entire image pixel by pixel with “*G*”. We defined a luminance factor “*a*” to represent the light-to-dark ratio of the image. It is worth noting that in order to clarify the following statements, we define some variables: OTSU(•) indicates that the optimal threshold is obtained by the Nobuyuki Otsu method, and poll[•] indicates that images are polled, compared, and counted.

We calculated the luminance factor “*a*” as follows:(16)$$G=OTSU(F_{in})$$
(17)$$\left\{\begin{array}{c} B=poll[F_{in}\left(x,y\right)\geq G]\\ D=poll[F_{in}\left(x,y\right)<G] \end{array}\right.$$
(18)$$a=\frac{B}{D}$$
where $F_{in}$ is the input figure.

Second, the contrast conversion coefficient CT was obtained by designing a new nonlinear element function. The image was weighted to obtain the pixel value $F_{trans}\left(x,y\right)$ and the image $F_{trans}$ after the final transform. The specific mathematical expression is as follows:(19)$$CT=\frac{R*(1/a)}{T-F_{in}\left(x,y\right)+\varepsilon }$$
(20)$$F_{trans}\left(x,y\right)=CT*F_{in}\left(x,y\right)$$
where $F_{in}\left(x,y\right)$ is the pixel value of the image in a certain spatial coordinate; *R* is the average pixel value of the input figure; *T* is the maximum pixel value of the input figure; ε is an infinitesimal quantity to prevent the denominator from being zero.

Finally, the two image matrices $F_{trans}$ and $F_{in}$ were transformed by a linear difference to obtained the target feature $F_{tf}$, where *a* is the luminance factor:(21)$$F_{tf}=\left\{\begin{array}{c} F_{trans}-F_{in}*a,a\leq 1\\ F_{trans}-F_{in}/a,a>1 \end{array}\right.$$

The specific feature extraction process is shown in Figure 3.

### 3.3. Spatial-Domain Logic Filter

The pixel values of the fused image were corrected by a logical filter to reduce the artifacts as well as the color loss due to the Gibbs effect during the transform domain filtering process.

First, the final base layer and the final detail layer were fused by linear superposition and then converted into a matrix. Multiple iterations were subsequently conducted for pixel-wise comparisons of grayscale values. Ultimately, the optimized outcome was achieved by assigning appropriate grayscale values to the resultant image. The specific process is shown in Figure 4.

Below, we focus on the mathematical principles of the logical filter. Two 3 × 3 sub-blocks (*A* and *B*) were extracted from the initial fusion image (*F*) and the Y-channel image (*Y*), respectively. A difference matrix (*C*) was generated by the difference calculation for these two sub-blocks, and we calculated the mean difference ($C_{m}$) in that region. Next, $C_{m}$ was compared with the difference matrix, element-by-element, to generate a new matrix ($I_{sub}$). The following is the mathematical analysis.

First, we obtained the difference matrix (*C*).
(22)$$\left[\begin{array}{ccc} c_{1} & c_{2} & c_{3}\\ c_{4} & c_{5} & c_{6}\\ c_{7} & c_{8} & c_{9} \end{array}\right]=\left|\begin{array}{c} \left[\begin{array}{ccc} a_{1} & a_{2} & a_{3}\\ a_{4} & a_{5} & a_{6}\\ a_{7} & a_{8} & a_{9} \end{array}\right]-\left[\begin{array}{ccc} b_{1} & b_{2} & b_{3}\\ b_{4} & b_{5} & b_{6}\\ b_{7} & b_{8} & b_{9} \end{array}\right] \end{array}\right|$$

Then, we calculated the mean of the matrix *C* and defined it as the mean difference ($C_{m}$) of the two sub-blocks *A* and *B*
(23)$$C_{m}=\frac{1}{D^{2}}\sum_{n=0}^{D^{2}}\left(c_{n}\right)$$

The next step was to poll $C_{m}$ to compare the elements in the matrix *C* and make a logical judgment as shown in Formula (24) to adjust the elements in the matrix *C*. We named the optimized *C* matrix $I_{sub}$ and defined the above method as logical filtering.
(24)$$\left\{\begin{array}{c} c_{n}=a_{n},ifc_{n}\leq C_{m}\\ c_{n}=b_{n},ifc_{n}>C_{m} \end{array}\right.$$

Finally, the initial fusion image *F* was filtered by the logical filter with a step size of 1, and the optimized image $I_{Fused}$ was obtained by reconstructing the matrix $I_{sub}$.

Figure 5 shows in detail the part of the optimization algorithm that relates to the logical filtering of fusion results. It can be seen that the detailed texture of $I_{Fused}$ is richer than *F*, and the Gibbs phenomenon is eliminated to a great extent without losing the target edge.

To improve understanding, Algorithm Section 3.3 provides a pseudocode depiction of the proposed DDCTFuse method.

**Algorithm 1** DDCTFuse**Input:** *Infrared image (IR) and visible image (VIS);* **Step 1:** Color-space transformations  Convert RGB and VIS to YCbCr  (Y,Cb,Cr)=YCbCrtransformation(VIS), **Step 2:** Image decomposition  do: IR and *Y* are decomposed into high-frequency parts (HIR, HY) and a low-frequency parts (LIR, LY) by the adaptive high-pass filtering.  (HIR,LIR)=BWfilter(IR),  (HY,LY)=BWfilter(Y),  then: Image detail retention of the high-frequency parts (HIR, HY) is performed using BF (bilateral filtering) to obtain NHIR and NHY.  (NHIR)=BFfilter(IR),  (NHY)=BFfilter(Y), **Step 3:** Contrast transformation  The contrast-transformed features are extracted from IR by the proposed contrast transform function to obtain the feature map NIR.  NIR=ContrastTransform(IR), **Step 4:** High-frequency fusion method  NIR is utilized to guide the fusion of the high-frequency parts (NHIR, NHY) to obtain the final detail layer (FDL).  FDL=concat(NHIR,NHY,NIR), **Step 5:** Low-frequency fusion method  The low-frequency parts (LIR, LY) are wavelet-decomposed and fused with a weighted average using the wavelet weighted-average fusion strategy to obtain the final base layer (FBL).  FBL=wavelet(LIR,LY), **Step 6:** Image fusion  FBL and FDL are fused to obtain the fusion result (IF).  IF=concat(FBL,FDL), **Step 7:** Optimization of fusion results  The fused image is optimized using the designed logical filtering optimization strategy based on a spatial-domain grayscale comparison to obtain the final result (FIF).  FIF=Optimization(IF), **Step 8:** Inverse color-space conversion  The fused image (FI) is obtained by using the inverse YCbCr transform on FIF.  FI=InverseYCbCr(FIF,Cb,Cr),**Output:** Fused image (FI);

## 4. Experiments and Analysis

### 4.1. Experimental Setup

The DDCTFuse method proposed in this paper was built in the MATLAB R2023a software environment, and all experiments were performed on the same hardware platform (NVIDIA GeForce RTX 4060Ti GPU, i5-13490F CPU) and software platform (Window11 operating system). In order to verify the superiority of DDCTFuse, nine infrared and visible image fusion methods were selected for comparison, including ADF [27], LatLRR [28], DenseFuse [29], IFCNN [30], GFF, DIVFusion [31], NestFuse [32], RFN-Nest [33], and SeAFusion [34].

The test datasets were LLVIP [35], MSRS [36], INO, and Roadscene [37]. LLVIP contains 12,025 groups of IR and visible images. MSRS contains 361 groups of IR and visible images. INO contains 55 groups of IR and visible images. EN, AG, MI, VIF, Nabf, and FMI were used as objective evaluation indexes of the fused image quality. To demonstrate the advantages of our adaptive high-pass filtering, based on the contrast transformation of nonlinear unit function, spatial-domain logic filtering, and low-frequency partial fusion strategy, we devised three sets of ablation experiments. In addition, we conducted real-time testing experiments to demonstrate the important development prospects of DDCTFuse.

### 4.2. Subjective Evaluation

To illustrate the advantages of DDCTFuse, we compared it with nine different classical algorithms on three different datasets. In order to show some of the image details, the infrared target part in the following figure is marked with a zoomed-in red box, and the background detail information is marked with a zoomed-in green box.

In Figure 6, SeAFusion, Nestfuse, and IFCNN have a strong ability to extract the infrared energy information in the source images, but the road texture cannot be accurately extracted in low-light conditions. Although the results of GFF retain some background information of the source visible image, the infrared target has obvious Gibbs artifacts and edge loss. DIVFusion, ADF, DenseFuse, RFN-Nest, and LatLRR demonstrate a heightened focus on global information, which leads to the inconspicuous infrared target. In comparison, our algorithm’s results possess a very significant and high-quality infrared target while preserving the source’s visible background details as well as color information.

As shown in Figure 7, ADF, DenseFuse, GFF, LatLRR, and RFN-Nest give a parallel level of attention to information in both source images, which creates a dim fused image. IFCNN, DIVFusion, and Nestfuse put more emphasis on preserving the color information of the visible image, and clearly extract the details of the tree and the house, which leads to some loss of the infrared target and causes ringing artifacts. SeAFusion and DDCTFuse have distinct IR targets and retain the background information of the source visible image to a large extent, and our proposed method has a subtle advantage in color preservation, as shown in the green-box enlargement section.

In Figure 8, GFF, ADF, IFCNN, DenseFuse, and LatLRR render enough attention to the global information of both source images, which result in the fused image possessing segmental background information of the infrared image. DIVFusion and RFN-Nest can well extract the contrast information of the visible image; in other words, it leads to the loss of a fraction of the infrared energy information. Nestfuse, SeAFusion, and DDCTFuse have a natural fusion effect, and the infrared target is prominent; moreover, the texture of the tree is relatively clear.

In Figure 9, since the visible image itself is blurred and of low quality, it results in RFN-Nest, DIVFusion, and DenseFuse focusing on the global features of the visible image, therefore presenting a misty fused image. GFF, IFCNN, ADF, Nestfuse, and SeAFusion have a strong ability to extract infrared energy information. However, they are unable to pay close attention to the edge information, which leads to artifacts. LatLRR and our method possess clear fusion results and significant infrared targets, and even the branches have rich detailed information.

### 4.3. Objective Evaluation

A subjective evaluation has a certain degree of one-sidedness and is susceptible to human factors. Therefore, we chose the following objective evaluation metrics: entropy (EN), average gradient (AG), mutual information (MI), No-reference assessment based on blur and noise factors (Nabf), Feature Mutual Information (FMI), and visual information fidelity (VIF) for our analysis. In total, 80 images from the MSRS dataset, 100 images from the LLVIP dataset, and all images in the Roadscene and INO datasets were selected for testing. In this case, except for Nabf, which takes a smaller value, a larger value is better for the other five objective evaluation indicators. The definitions of each metric are as follows:

(1) The information entropy (EN) serves as an objective assessment metric for quantifying the amount of information present in an image. In this context, “*a*” represents the grayscale value, while P(a) signifies the probability distribution of gray levels. A higher EN value indicates a greater abundance of information and superior quality in the fused image.
(25)$$H\left(A\right)\;=\;-\sum_{a}P_{A}\left(a\right)logp_{A}\left(a\right)$$

(2) The clarity of the fused image can be evaluated using the average gradient (AG), where a higher average gradient indicates superior image clarity and fusion quality. The formula for computing AG is as follows:(26)$$AG\;=\;\frac{1}{\left(M-1)(N-1\right)}\;\sum_{i=1}^{M-1}\sum_{i=1}^{N-1}\sqrt{\frac{{\left(H(i+1,j)-H(i,j)\right)}^{2}+{\left(H(i,j+1)-H(i,j)\right)}^{2}}{2}}$$
where *H* is the composite image, with *M* and *N* representing its dimensions in terms of height and width, respectively.

(3) The mutual information (MI) serves as a quantification of the degree of similarity between two images, indicating the extent to which information from the original image is preserved in the fused image. A higher mutual information signifies superior fusion quality by retaining more information from the source image. The computation of mutual information involves considering both individual images’ entropy H(A) and their joint entropy H(A,B).
(27)MI(A,B)=H(A)+H(B)−H(A,B)

(4) The complete designation of Nabf is “No-reference evaluation relying on factors of blur and noise”, indicating its function as a no-reference method for assessing image quality based on the presence of blur and noise. It accurately determines the degree of image blurriness and noise without requiring a reference image for comparison.

(5) The fidelity of information is commonly assessed using the visual information fidelity (VIF) metric, which serves as a measure for evaluating the subjective visualization quality in images. A higher VIF value indicates a superior visual representation. The calculation is executed through a four-step process, accompanied by the following simplified formula:(28)$$VIF=\frac{VID}{VIND}$$
where the acronyms VID and VIND denote the visual information derived from the composite image generated by the input image.

(6) Feature Mutual Information (FMI) is a method for measuring the similarity between two images, which is based on feature information theory and mutual information theory. The similarity between two images is quantified by a metric that measures the occurrence of shared pairs of feature points. The formula provided below presents a straightforward approach for evaluating this aspect:(29)$$FMI_{F}^{AB}=\frac{I_{FA}}{H_{F}+H_{A}}+\frac{I_{FB}}{H_{F}+H_{B}}$$
where *F* denotes the fused image, and *A*, *B* are the source images; $I_{FA}$ and $I_{Fb}$ are the feature information of *A* and *B* contained in *F* represented through MI; the entropy values based on the histograms of images *A*, *B*, and *F* are represented by HA, HB, and HF, respectively.

In Table 1, the average values of 80 randomly selected sets of image evaluation metrics in the MSRS dataset are shown, where the optimal values are marked in bold, the second best in red, and the third in blue.

From the objective evaluation metrics in the above table, it can be seen that the proposed algorithm occupies four optimal solutions and one suboptimal result, which proves that the fused image contains more detail information and has high fusion quality and great contrast.

We selected one hundred images from the LLVIP dataset for testing, and our method performed well, with optimal values for AG, MI, Nabf, and FMI, as shown in Table 2. This substantiates DDCTFuse’s advantages and underscores its superior robustness.

The Roadscene dataset was tested, and all 221 images were examined. The comparison results between DDCTFuse and nine other image fusion algorithms on the Roadscene dataset are presented in Table 3, where MI, VIF, FMI, and Nabf had optimal values. The results of the objective evaluation metrics demonstrate that DDCTFuse effectively preserves the intricate details of the source images, exhibits strong consistency with subjective perception, and yields high-quality fused images.

Shown in Table 4 are the test results of all algorithms on the INO dataset. Each algorithm had its own advantages. DDCTFuse obtained the best MI and Nabf values, which indicates that the fused image can better retain the information of the source image.

### 4.4. Ablation Experiments

This section aims to further showcase the advantages of our adaptive high-pass filter, contrast transform feature extraction, and logical filtering optimization algorithms. Due to space constraints, we conducted three sets of ablation experiments on the LLVIP and MSRS datasets and analyzed the results.

The first set of ablation experiments aimed to highlight the benefits of the adaptive high-pass filter in the frequency-domain image decomposition. We replaced the adaptive high-pass filter with two commonly used high-pass filters: a traditional Gaussian high-pass filter and an ideal filter. The specific results are presented in Figure 10, Figure 11 and Figure 12.

The experimental results clearly demonstrated the effectiveness of our proposed adaptive high-pass filter in accurately extracting both high- and low-frequency components from images. Compared to other prevalent high-pass filters, our innovative approach generated a more comprehensive and distinctive depiction of the target’s high-frequency details, thereby enhancing overall accuracy and effectiveness in image processing tasks.

The adaptability of our proposed high-pass filter was particularly crucial when dealing with images that had significant variations in their characteristics. By dynamically adjusting the filter parameters based on each image’s specific features, our approach ensured accuracy in extracting both high- and low-frequency components, even when faced with complex or challenging image content.

Furthermore, our adaptive high-pass filter excelled at preserving the integrity of the target image’s important high-frequency details, which are essential for accurate interpretation and analysis of visual information.

In order to prove the outstanding performance of contrast transform feature extraction and logical filtering optimization in nonlinear fusion in the spatial-domain, a second set of ablation experiments was designed as follows: deletion of the contrast transform feature extraction, deletion of the logical filtering optimization in the spatial-domain, deletion of both features. The specific results are presented in Figure 13 and Figure 14.

In the set of figures presented above, we have four distinct outputs that serve to illustrate the effectiveness of our proposed method for fusing infrared images.

The outputs of DDCTFuse in column (a) serve as a reference for evaluating the effectiveness of the other three methods.

The results of removing the contrast transform feature extraction part are depicted in column (b). These outputs offer valuable insights into the significance of feature extraction in the fusion process, as evidenced by the comparatively diminished effectiveness of detecting the infrared target in the fused image when compared to our proposed method. This outcome further substantiates the superior performance of our proposed method, highlighting its superiority over other techniques in terms of fusing infrared images.

The outputs of the removal of the spatial-domain logistic filter are examined in column (c). The fused image appears relatively darker compared to the others and exhibits a certain degree of color distortion. This observation suggests that the spatial-domain logistic filter plays a crucial role in achieving intensity balance and ensuring color accuracy in the fused image.

The outputs in column (d) demonstrate the results of removing both the feature extraction component of the contrast transform and the spatial-domain logistic filter. In this case, the fused image appears significantly darker and exhibits a more pronounced degree of color distortion, thereby emphasizing the crucial role played by these elements in the fusion process.

Overall, these outputs from four sets of experiments serve to demonstrate the effectiveness of our proposed method and the importance of each component in the fusion process. The comparison between these outputs and our proposed method highlights the direct effectiveness of our approach, making it a promising technique for fusing infrared images.

In order to demonstrate the effectiveness of the weighted wavelet transform in the low-frequency part fusion, a third set of ablation experiments was conducted using a direct fusion strategy and a linear average fusion strategy for comparison. The specific subjective and objective output results are shown below.

In Figure 15, the infrared target is indicated by the red boxes, and it can be observed that the direct fusion algorithm yielded edge blurring in its result. The background information highlighted in green boxes demonstrates that the linear averaging algorithm and the direct fusion algorithm had information loss.

In order to further show the advantages of the weighted-wavelet fusion algorithm in information retention, we use ten evaluation indicators for objective tests: EN, AG, MI, VIF, Nabf, FMI, SF, SD, Qabf, and CC. The resulting output was as follows. Due to limited space, only two decimal places were retained. Moreover, the direct fusion is referred to as DF, and the linear average fusion is referred to as LF.

As illustrated in Table 5, the optimal values, namely, EN, AG, MI, Nabf, FMI, SF, and Qabf, significantly emphasize the efficacy of the wavelet-weighted fusion strategy for the low-frequency part fusion. This strategy not only effectively preserved the source image information but also generated a discernible fusion outcome, capturing minute visual details from the input image. Furthermore, the strategy enhanced the overall quality of the fused image by incorporating the wavelet transform, which inherently focused on the low-frequency components of the image. Consequently, the fused image exhibited a higher level of detail and a more accurate representation of the original source images, demonstrating the superiority of the wavelet-weighted fusion strategy over other fusion methods. To visualize the difference between the results of the various methods, we present a histogram in Figure 16.

### 4.5. Real-Time Validation Experiment

In order to verify that DDCTFuse had good visual output effect and real-time performance, we compared DDCTFuse with some current advanced image fusion algorithms (RFN-Net, DenseFuse, NestFuse, and LatLRR) on the LLVIP and MSRS datasets, mainly in terms of CPU or GPU usage, RAM usage, and algorithm response time. The specific experimental results are shown in Table 6. It should be noted that the above-mentioned deep learning methods all completed the model training and reasoning process in the same virtual environment (python-3.8.10; torch-2.2.0+cu121; torchvision-0.17.0). The specific experimental results are shown in Table 6.

From the above experimental results, it can be concluded that the DDCTFuse method proposed in this paper achieved good performance in real time. When DDCTFuse was running, the CPU usage and the RAM usage was minimal, the algorithm response time was short, and the output result was fast. The above advantages prove that DDCTFuse is very advantageous in actual application scenarios.

## 5. Discussion

Combining the subjective and objective evaluations in the previous section and the results of the ablation experiments, we proved that the new contrast transform function we designed had excellent performance in the IR target extraction task, which can be reflected by the value of VIF and FMI. The spatial-domain logic optimized filter proposed in this paper also played an equally important role, as can be seen by the larger MI and AG.

Looking ahead, we plan to further refine the nonlinear element function for the contrast transform and extend its application to other computer vision tasks such as edge detection and image segmentation. We anticipate that this expansion will not only enhance the performance of these tasks but also broaden the scope of our research. Additionally, we are considering the integration of the logical filter into deep learning methods for color correction at various feature levels. This interdisciplinary approach holds the potential to make substantial advancements in the field of computer vision and open new avenues for research and development.

## 6. Conclusions

In this paper, we proposed an infrared and visible image fusion strategy (DDCTFuse) based on an adaptive high-pass filter, a logical filter, and contrast transform feature extraction, which enhanced the infrared target and transformed problems that are difficult to deal with in the frequency domain, such as ringing artifacts, into the spatial-domain by using a spatial-domain logical filter for processing. The experimental results showed that DDCTFuse achieved superior visual effects and evaluation results with strong generalization ability and good robustness when compared with nine fusion algorithms on four public datasets.

## Figures and Tables

**Figure 1 sensors-24-03949-f001:**
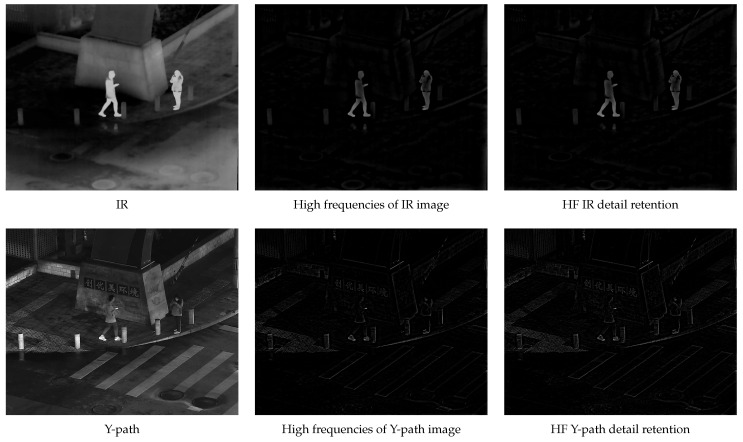
BF detail retention.

**Figure 2 sensors-24-03949-f002:**
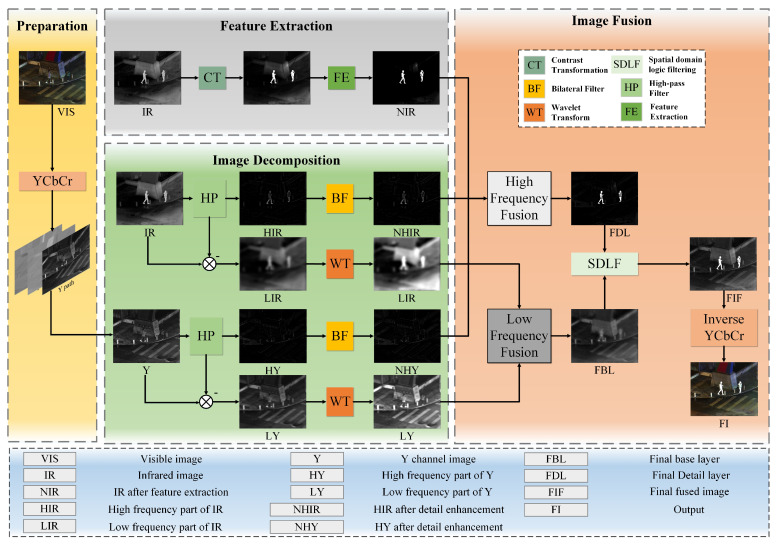
Schematic of the proposed DDCTFuse method: the gray dotted line box illustrates the specific algorithmic steps, the blue box presents the full image name, the black dotted line box presents the complete names of distinct algorithms within the fusion strategy, facilitating readability.

**Figure 3 sensors-24-03949-f003:**
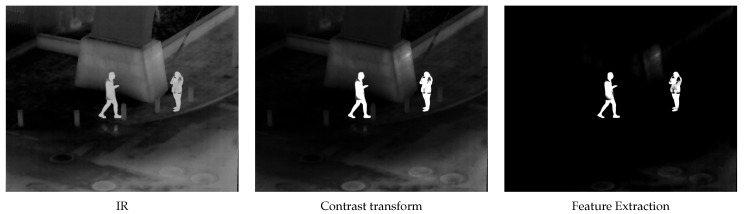
Contrast transform feature extraction process.

**Figure 4 sensors-24-03949-f004:**
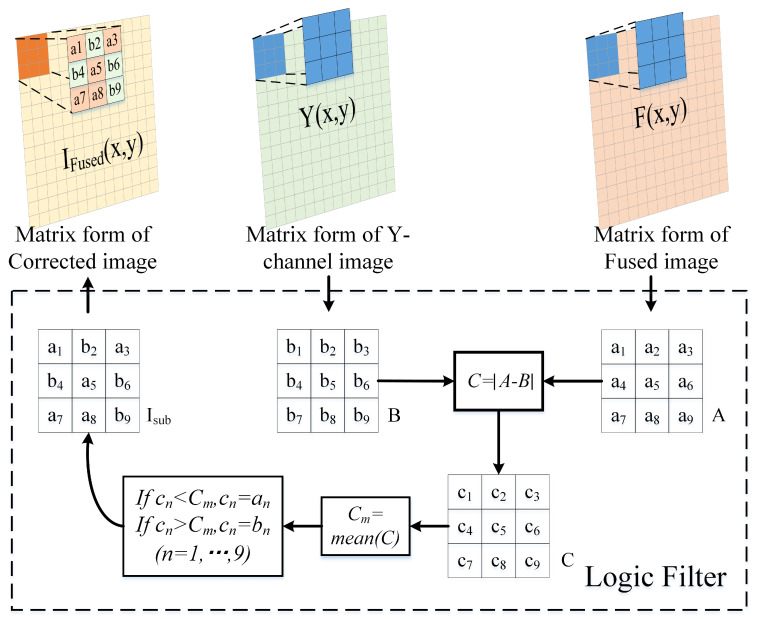
Logic filtering process: the elements of *F*, denoted as F(x,y), represents the pixel value of *F*. $a_{n},\left(n=1,...,9\right)$ denotes the pixel value in sub−block A. The pixel value of the Y−channel image is represented by Y(x,y). $b_{n},\left(n=1,...,9\right)$ denotes the pixel value in sub−block B. *C* represents the difference matrix.

**Figure 5 sensors-24-03949-f005:**
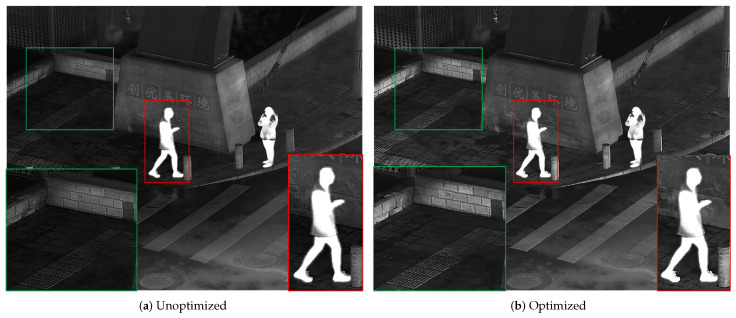
Comparison of logic filter optimization results. The infrared target edge details have been marked in red boxes and the background details have been marked in green boxes.

**Figure 6 sensors-24-03949-f006:**
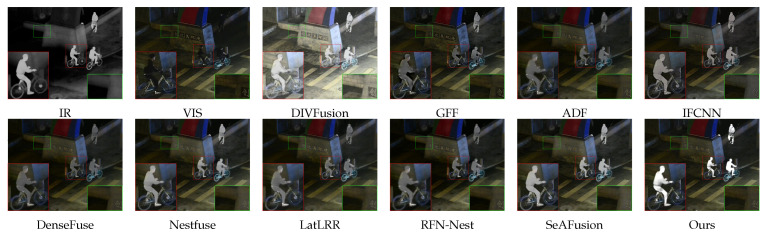
Comparison results on “010370” in the LLVIP dataset.

**Figure 7 sensors-24-03949-f007:**
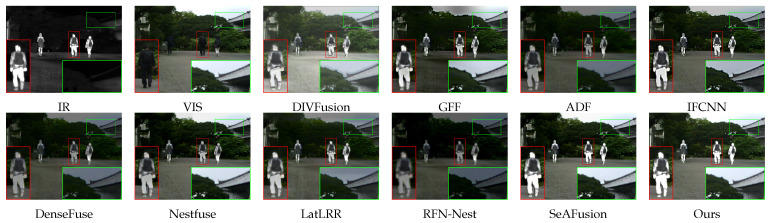
Comparison results on “00634D” in the MSRS dataset.

**Figure 8 sensors-24-03949-f008:**
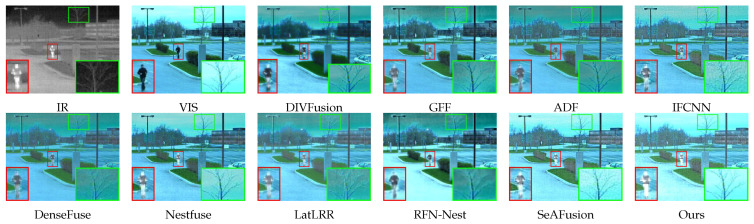
Comparison results on “44” in the INO dataset.

**Figure 9 sensors-24-03949-f009:**
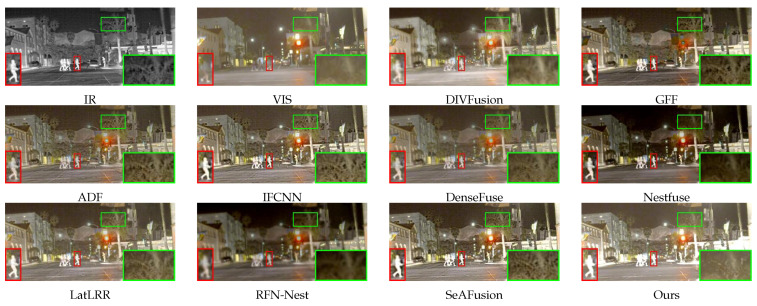
Comparison results on “FLIR-09016” in the Roadscene dataset.

**Figure 10 sensors-24-03949-f010:**
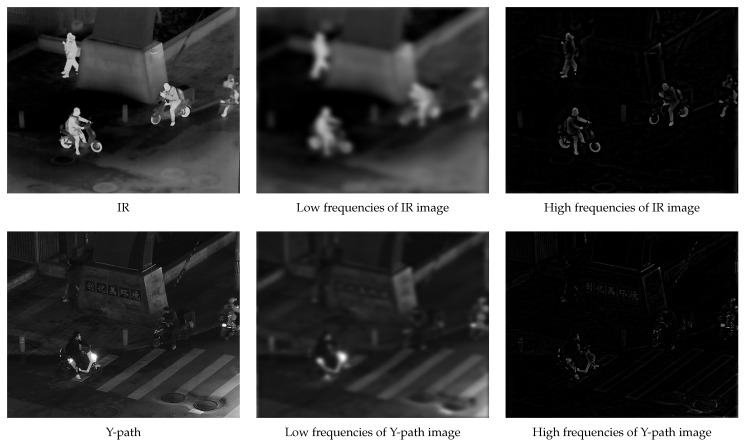
Adaptive high-pass filtering frequency-domain decomposition results.

**Figure 11 sensors-24-03949-f011:**
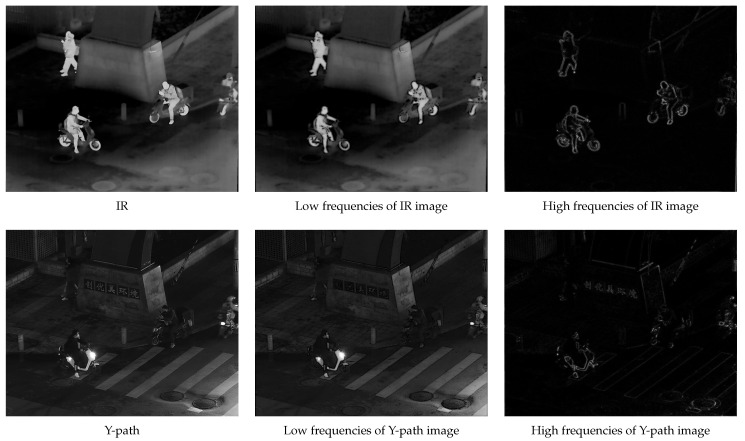
Gaussian high-pass filter frequency-domain decomposition results.

**Figure 12 sensors-24-03949-f012:**
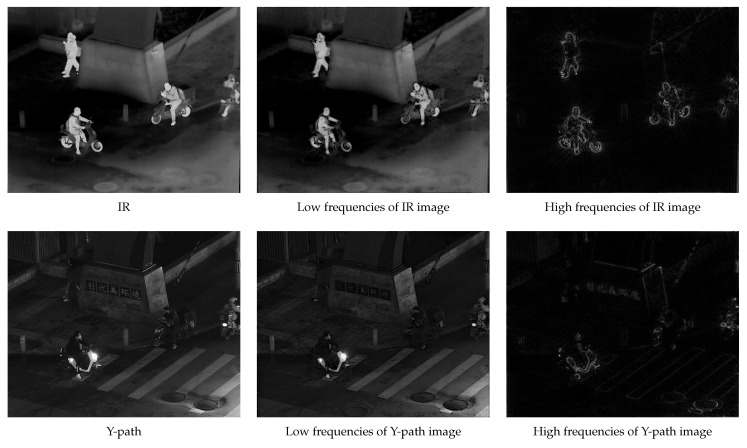
Ideal high-pass filter frequency-domain decomposition result.

**Figure 13 sensors-24-03949-f013:**
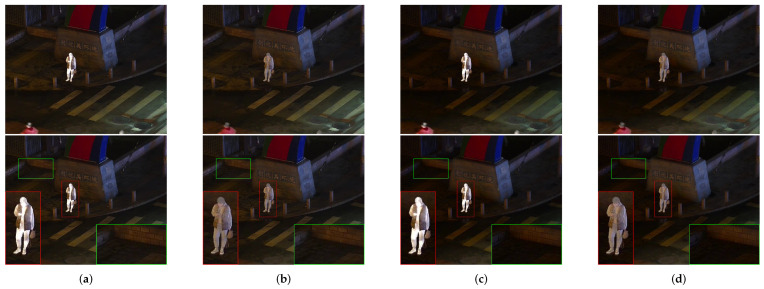
Ablation experiments on “010037” in the LLVIP dataset. The infrared target edge details have been marked in red boxes and the background details have been marked in green boxes.

**Figure 14 sensors-24-03949-f014:**
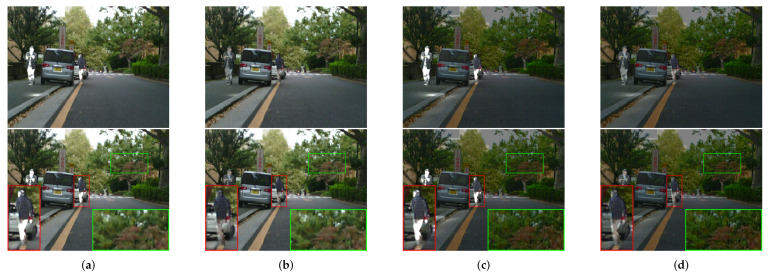
Ablation experiments on “00613D” in the MSRS dataset. The infrared target edge details have been marked in red boxes and the background details have been marked in green boxes.

**Figure 15 sensors-24-03949-f015:**
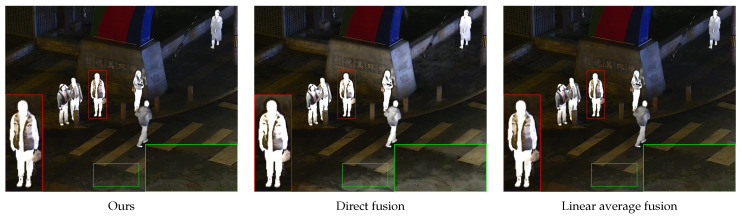
Subjective results of ablation experiments. The infrared target edge details have been marked in red boxes and the background details have been marked in green boxes.

**Figure 16 sensors-24-03949-f016:**
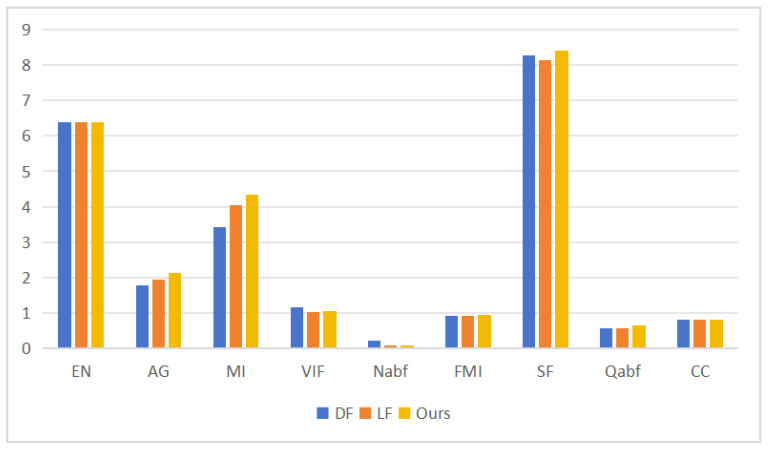
Histogram of the third set of objective evaluation indicators from the ablation experiment.

**Table 1 sensors-24-03949-t001:** Objective evaluation metrics of 9 comparison algorithms and our proposed algorithm on the MSRS dataset.

Algorithm	EN	AG	MI	VIF	Nabf	FMI
ADF	5.9054	2.6496	2.4144	0.7080	0.0605	0.9106
DenseFuse	5.8657	2.1403	2.4066	0.6675	0.0504	0.9193
DIVFusion	**7.4803**	**4.5438**	2.4721	0.7668	0.3079	0.8799
GFF	6.2610	3.3391	2.2872	0.8714	0.0773	0.9323
IFCNN	6.3944	4.0352	2.6651	0.8124	0.2364	0.9191
LatLRR	6.2447	2.7243	2.2384	0.7826	0.0759	0.9200
NestFuse	6.0739	3.3290	3.4932	0.8679	0.1438	0.9261
RFN-Nest	5.4312	2.0016	2.1027	0.6012	0.0496	0.9189
SeAFusion	6.4042	3.8455	3.4686	0.9401	0.2155	0.9263
Ours	6.5959	3.3702	**5.8577**	**0.9695**	**0.0460**	**0.9347**

**Table 2 sensors-24-03949-t002:** Objective evaluation metrics of 9 comparison algorithms and our proposed algorithm on the LLVIP dataset.

Algorithm	EN	AG	MI	VIF	Nabf	FMI
ADF	6.3888	1.7793	2.6872	0.8978	0.1069	0.9403
DenseFuse	6.3785	1.5676	2.7953	0.8989	0.08522	0.9460
DIVFsuion	**7.4952**	**4.5587**	2.5919	0.7943	0.3264	0.9033
GFF	6.3560	2.1034	2.6164	1.1334	0.1186	0.9411
IFCNN	6.6042	2.1479	3.0449	0.9823	0.1540	0.9396
LatLRR	6.4300	1.4789	2.5881	0.8466	0.08545	0.9363
NestFuse	6.7731	1.6542	3.5813	1.0986	0.0842	0.9413
RFN-Nest	6.5803	1.3783	2.8061	0.9908	0.08539	0.9324
SeAFusion	6.5947	2.0777	3.6510	**1.1364**	0.1380	0.9441
Ours	6.6521	2.1506	**4.2829**	1.0526	**0.0834**	**0.9671**

**Table 3 sensors-24-03949-t003:** Objective evaluation metrics of 9 comparison algorithms and our proposed algorithm on the Roadscene dataset.

Algorithm	EN	AG	MI	VIF	Nabf	FMI
ADF	7.0962	6.0353	1.0238	0.0802	0.7495	0.3788
DenseFuse	6.9735	3.7808	1.1083	0.0857	0.7708	0.2779
DIVFsuion	7.5506	4.7561	2.8942	0.7129	0.8429	0.1427
GFF	7.5728	5.0169	3.3935	0.0963	0.8024	0.1764
IFCNN	7.2164	6.1860	1.0298	0.0789	0.7627	0.3937
LatLRR	7.0560	4.0818	1.1083	0.0883	0.7707	0.2871
NestFuse	**7.5735**	5.6342	1.2769	0.0916	0.7734	0.3490
RFN-Nest	7.4774	3.7637	1.3288	0.1003	0.7914	0.2785
SeAFusion	7.4850	**7.2913**	1.2021	0.0863	0.7642	0.4269
ours	6.9131	5.9633	**5.6008**	**0.7763**	**0.8695**	**0.1331**

**Table 4 sensors-24-03949-t004:** Objective evaluation metrics of 9 comparison algorithms and our proposed algorithm on the INO Dataset.

Algorithm	EN	AG	MI	VIF	Nabf	FMI
ADF	6.8171	5.2523	2.9866	0.7244	0.8301	**0.0655**
DenseFuse	6.7553	4.2104	2.9034	0.7126	0.8557	0.0842
DIVFsuion	**7.4305**	5.7465	3.2211	0.8111	0.8468	0.1529
GFF	7.1982	6.5936	3.6897	**0.9121**	0.8847	0.0773
IFCNN	6.8974	7.5545	2.4015	0.7096	0.8599	0.2811
LatLRR	6.8863	5.0988	2.6903	0.6749	0.8476	0.1293
NestFuse	7.0922	6.6856	2.8999	0.8155	0.8549	0.1966
RFN-Nest	7.3883	5.8763	3.2622	0.7875	0.8605	0.1478
SeAFusion	7.1964	**8.1769**	2.7893	0.7744	0.8671	0.3223
Ours	6.8644	7.0469	**4.6567**	0.8726	**0.8906**	0.1873

**Table 5 sensors-24-03949-t005:** Objective evaluation metrics for the third set of ablation experiments on the LLVIP dataset.

Algo	EN	AG	MI	VIF	Nabf	FMI	SF	SD	Qabf	CC
DF	6.38	1.78	3.42	**1.16**	0.23	0.91	8.26	7.84	0.58	**0.82**
LF	6.37	1.95	4.03	1.03	**0.08**	0.93	8.12	**8.86**	0.56	0.80
Ours	**6.39**	**2.12**	**4.33**	1.06	**0.08**	**0.94**	**8.39**	7.87	**0.64**	0.81

**Table 6 sensors-24-03949-t006:** Performance comparison of algorithms on different datasets.

Dataset	Algorithm	CPU	GPU	RAM	Time (s)
	LatLRR	65%	—	51 MB	0.331
	RFN-Net	—	98%	3.5 GB	0.836
MSRS	DenseFuse	—	100%	0.7 GB	0.451
	NestFuse	—	96%	1.3 GB	0.764
	Ours	14%	—	32 MB	0.051
	LatLRR	47%	—	67 MB	0.384
	RFN-Net	—	100%	3.7 GB	0.913
LLVIP	DenseFuse	—	99%	0.8 GB	0.414
	NestFuse	—	99%	1.5 GB	0.815
	Ours	17%	—	34 MB	0.049

## Data Availability

The data presented in this study are available on request from the corresponding author due to privacy.

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
