# Peer review of "Infrared and Visible Image Fusion Algorithm Based on Double-Domain Transform Filter and Contrast Transform Feature Extraction"

_sensors, 2024, doi:10.3390/s24123949_

Round 1

Reviewer 1 Report

Comments and Suggestions for Authors

This paper has several flaws.

1.    The writing does not satisfy the technical standard. For example, the figure caption often contains several description words and then after a blank row starts with “where…..”. This is my first time to see such kind of technical writing.

2.   The English grammar needs to be further corrected.

3.   The technical part included is too basic. For examples, page 5 (from Eq. (5)-(13) for DFT spectrum computation), page 7 (from Eq.(14)-(18) for Otsu’s algorithm), page 3 (Eq.(1)-(4)). Basically, these three parts are very basic and don’t need to be included in an advanced paper/journal. Even, the Otsu’s algorithm for binary thresholding was published in 1979. I studied that 20 years ago and consider that the description about the Otsu’s algorithm here is still somehow wrong (e.g., intra-class variance is ignored). By the way, how B and D are chosen in Eq. (18) is not addressed. From the text, it is not clear where to apply the bilateral filter. Finally, by searching the keyword and I found it in Algorithm 1 (page 10). Algorithm 1 is only provided as a summary for helping the readers. The use of bilateral filter should be mentioned in the text.

4.   The technical description is not clear, inconsistent, and sometime erroneous. For examples, in Algorithm 1, it describes “Bilateral filtering is used to derive NHIR and NHY. However, in Fig. 2, it is “DE (Detail Enhancement” that leads to NHIR and NHY. Actually, I don’t think bilateral filter can enhance the details. BL is one kind of smoothing filter that is capable of preserving edges only.

5.   The adaptive high-pass filter chooses the mean frequency (line 162) as the cutoff frequency, as illustrate in Eq.(11). However, Eq.(11) is wrong. Eq.(11) computes the average amplitude of the spectrum. However, D0 is the cutoff frequency. Hence, the description in Eq.(11) is wrong. Actually, the choice of the mean frequency (the frequency which results in the mean amplitude) as the cutoff frequency is also very common since traditionally, 2dB is often used as the criterion as the cutoff frequency. The comparison of D0=15.348 with 1 (very small) and 40 (very large) to show the advantage is not proper.

6.   The description from Eq.(19)-(24) is somehow unclear and erroneous. For example, b function is not defined and b duplicates with k function (in Eq.(21)). In the whole manuscript, D are repeatedly defined for different quantities (at least 4 times). This needs to be prevented. I also think that there are several kinds of contrast transform (like histogram equalization, local contrast enhancement, contrast stretching) which can be used to increase the contrast. Hence, the proposal of Eq.(21) for contrast transform is not significant. In Eq.(23), the coefficient b is also not defined. In Eq.(24), it is impossible for a to be negative since a is calculated from Eq.(22). In line 228, “B and D represent the total number of grayscale values in the bright and dark parts”. This statement is not consistent with Eq.(18) where the image is binarized into values B and D only.

7.   Section 2.3 for spatial domain logic filter is rather unclear. First of all, it is not clear what image it is operated on? Is it FDL+FBL in Fig. 2? It is not clear how to fuse FDL and FBL (by addition?). This is also not consistent with F(x,y) in Fig. 5, which is also not defined. The authors often use “filter kernels” in Section 2.3. This is not correct. It should be “sub-image covered by the filter kernel”. It is also inconsistent between Fig. 2 and Fig. 5. For example, in Fig. 5, it presents that SDLF is operated on F and Y, however, in Fig. 2, it presents that SDLF is operated on FDL and FBL. I also think that C should be named as “difference matrix”, but not “error matrix”. The writing of Eq.(27) is also not standard. It should be “cn=an if cn < Cm”, but not “cn < Cm, cn=an”. In line 260, “convolution kernel A” is also not proper since the operation in SDLF is not convolution.

8.   There are several repeated LLVIP in captions of Table 3 and 4. It’s wrong.

Comments on the Quality of English Language

Needs to be further improved though it is generally understandable.

Reviewer 2 Report

Comments and Suggestions for Authors

The paper titled “Infrared and visible image fusion algorithm based on double domain transform filter and contrast transform feature extraction”. The authors have to address the following before the paper accepted:

  1. Lack of Detailed Explanation of Methods: The authors could provide a more in-depth explanation of the methods used, such as the adaptive high-pass filter, logical filter, and contrast transform feature extraction. This could include diagrams, pseudocode, or a step-by-step walkthrough of the process.

  2. Insufficient Analysis of Results: The authors could include specific examples or data points from the experiments to support their claims. This could involve detailed tables, graphs, or statistical analysis comparing their method with others.

  3. Limited Discussion on Future Research: The authors could outline more concrete plans or hypotheses for future research. This could include potential improvements to the contrast transform function, or ways to integrate the logical filter into deep learning methods.

  4. Superficial Reference to Ablation Studies: The authors could provide more detailed information about the specific ablation studies conducted and their outcomes. This could help readers understand the effectiveness of each component of the proposed method.

  5. References Formatting and Completeness: The authors should ensure that all references are complete and uniformly formatted. This is crucial for the credibility of the paper.
  6. General Flow and Readability: The authors could improve the overall readability of the paper by ensuring a logical flow between sections and avoiding abrupt transitions. This could involve integrating the discussion of methods, results, and future work more seamlessly.
Comments on the Quality of English Language

The paper titled “Infrared and visible image fusion algorithm based on double domain transform filter and contrast transform feature extraction”. The authors have to address the following before the paper accepted:

  1. Lack of Detailed Explanation of Methods: The authors could provide a more in-depth explanation of the methods used, such as the adaptive high-pass filter, logical filter, and contrast transform feature extraction. This could include diagrams, pseudocode, or a step-by-step walkthrough of the process.

  2. Insufficient Analysis of Results: The authors could include specific examples or data points from the experiments to support their claims. This could involve detailed tables, graphs, or statistical analysis comparing their method with others.

  3. Limited Discussion on Future Research: The authors could outline more concrete plans or hypotheses for future research. This could include potential improvements to the contrast transform function, or ways to integrate the logical filter into deep learning methods.

  4. Superficial Reference to Ablation Studies: The authors could provide more detailed information about the specific ablation studies conducted and their outcomes. This could help readers understand the effectiveness of each component of the proposed method.

  5. References Formatting and Completeness: The authors should ensure that all references are complete and uniformly formatted. This is crucial for the credibility of the paper.
  6. General Flow and Readability: The authors could improve the overall readability of the paper by ensuring a logical flow between sections and avoiding abrupt transitions. This could involve integrating the discussion of methods, results, and future work more seamlessly.

Reviewer 3 Report

Comments and Suggestions for Authors

It is a useful research project with high accuracy results. However, I have two questions as follows

1) What about CPU or GPU usages when doing inference?

2) It should be identified infrared images on real-time usages. How much time does algorithm need?

I would recommend that your team add these experimental data.

Comments on the Quality of English Language

English language is good enough for understanding.

Round 2

Reviewer 1 Report

Comments and Suggestions for Authors

1.      Line 178, A(u,v) should be corrected to be F(u,v).

2.      Eq.(10)(11) are wrong. The right most u0 and v0 in the numerators should be replaced with, e.g., the horizontal and vertical bandwidths, respectively.

3.      Line 197. I don’t understand what is “inter-law variance”? Also, there is an inconsistency between the full name and the abbreviation name for “maximum inter-law variance method (OTSU)”.

4.      Line 232, I_sub must be clearly expressed. But we cannot find it in Eq.(22)-(24).

5.      Line 430. Figure (a), which figure? Figure 13(a) and Figure 14(a)? Similarly for Line 433, 438, 443.

Comments on the Quality of English Language

acceptable
